# A Randomized Comparison of Plasma Levobupivacaine Concentrations Following Thoracic Epidural Analgesia and Subpleural Paravertebral Analgesia in Open Thoracic Surgery

**DOI:** 10.3390/jcm9051395

**Published:** 2020-05-09

**Authors:** Jan Matek, Stanislav Cernohorsky, Stanislav Trca, Zdenek Krska, David Hoskovec, Jan Bruthans, Martin Sima, Pavel Michalek

**Affiliations:** 11st Department of Surgery—Department of Abdominal, Thoracic Surgery and Traumatology, First Faculty of Medicine, Charles University in Prague and General University Hospital in Prague, 128 00 Prague, Czech Republic; jan.matek@vfn.cz (J.M.); stanislav.cernohorsky@vfn.cz (S.C.); stanislav.trca@vfn.cz (S.T.); zdenek.krska@vfn.cz (Z.K.); david.hoskovec@vfn.cz (D.H.); 2Medical Faculty, Masaryk University, 625 00 Brno, Czech Republic; 3Department of Anaesthesia and Intensive Care, First Faculty of Medicine, Charles University in Prague and General University Hospital in Prague, 128 00 Prague, Czech Republic; jan.bruthans@vfn.cz; 4Institute of Pharmacology, First Faculty of Medicine, Charles University in Prague and General University Hospital in Prague, 128 00 Prague, Czech Republic; martin.sima@vfn.cz; 5Department of Anaesthesia, Antrim Area Hospital, Antrim BT41 2RL, UK

**Keywords:** thoracotomy, epidural analgesia, subpleural catheter, levobupivacaine, toxicity

## Abstract

Background: The aim of this study was to compare plasma levobupivacaine concentrations in thoracic epidural and subpleural paravertebral analgesia. Methods: Forty-four patients indicated for open lung resection had an epidural catheter inserted preoperatively or a subpleural catheter surgically. A bolus of 0.25% levobupivacaine at a dosage of 0.5 mg × kg^−1^ was given after the thoracotomy closure. Plasma levobupivacaine level at 30 min was the primary outcome. Pharmacokinetic modeling was performed subsequently. Secondary outcomes included the quality of analgesia, complications, and patients’mobility. Results: Plasma concentrations were similar 30 min after application—0.389 mg × L^−1^ in the epidural and 0.318 mg × L^−1^ in the subpleural group (*p* = 0.33) and lower in the subpleural group at 120 min (*p* = 0.03). The areas under the curve but not maximum concentrations were lower in the subpleural group. The time to reach maximum plasma level was similar in both groups—27.6 vs. 24.2 min. No clinical symptoms of local anesthetic toxicity were recorded. Conclusions: Levobupivacaine systemic concentrations were low in both groups without the symptoms of toxicity. This dosage should be safe for postoperative analgesia after thoracotomy.

## 1. Introduction

Lateral thoracotomy is generally associated with moderate to severe pain in the postoperative period and insufficient postoperative analgesia may result in worsening of postoperative outcomes including respiratory parameters, pulmonary infection and length of hospital stay [1]. Regional anesthetic techniques are generally considered as superior to systemic opioid analgesia and patients without regional anesthesia have a higher risk of severe pain intensity in the early postoperative period [2]. Several methods of providing postoperative analgesia following open thoracic surgery have been used in clinical practice. Continuous thoracic epidural administration of local anesthetics or in combination with a strong opioid or other additives has been considered a method of choice for many decades [3]. Other methods of regional analgesia, such as continuous percutaneous paravertebral block, subpleural paravertebral analgesia, interpleural administration, intrathecal application of morphine [4] or continuous erector spinae block [5] have been proposed as alternatives.

Continuous subpleural paravertebral analgesia performed by a surgeon under direct vision was introduced to clinical practice in 2012 as an alternative to thoracic epidural analgesia in patients who are not suitable for insertion of a thoracic epidural catheter [6]. Although probably less effective in providing pain relief, it can be beneficial in patients at high risk of side effects of an epidural block. There have been a few pharmacokinetic studies published assessing the pharmacokinetics of levobupivacaine in lumbar or lower thoracic epidural anesthesia or analgesia [7,8,9] and one study of paravertebral application [10]. Data about the pharmacokinetics of local anesthetic solutions following administration to the middle and upper thoracic epidural spaces or the subpleural catheters are not available.

The primary goal of this study was to compare the pharmacokinetics of the local anesthetic levobupivacaine after administration to the surgically inserted multi-holed subpleural paravertebral catheter versus the same drug administered to the thoracic epidural space to investigate whether there are any risks of potentially high levels of the drug.

## 2. Experimental Section

The study was designed as single-center, prospective and randomized. It was conducted at a Tertiary University Hospital. The study protocol was approved by the Local Ethical Committee (No. 1635/16 S-IV, 15/12/2016, Ethics Committee of the General University Hospital, Prague, chairperson Dr. J. Sedivy) and prospectively registered with the Australian and New Zealand Clinical Trial Register (ACTRN12616001541493). Following the signing of written informed consent, patients with American Society of Anesthesiologists (ASA) physical condition classes I–IV, older than 18 years and scheduled for elective open thoracic surgery (lobectomy or pneumonectomy) were considered for enrollment to the trial. The study ran between May 2017 and March 2018. Exclusion criteria included: age less than 18 years, acute or emergency surgery, contraindication to thoracic epidural or subpleural paravertebral analgesia, chronic pain prior to the surgical procedure and known allergy to levobupivacaine or sufentanil.

### 2.1. Randomization and Blinding

A list of random numbers was generated by the computer software (www.graphad.com) with allocation to the two groups—thoracic epidural analgesia (TEA) and subpleural paravertebral analgesia (SPA)—in a 1:1 ratio. The random numbers with the group codes were inserted in sealed envelopes which were then opened after patient enrollment on the morning of surgery. The assessors of postoperative variables (intensity of pain, mobility, and complications) were blinded to group allocation. The blinding was not feasible for the anesthesiologists or surgeons performing the procedures in the operating room. The patients were not informed about the result of randomization, however, the difference in timing of both techniques of regional anesthesia made them aware of group allocation.

### 2.2. Procedures

All patients received 0.25–0.5 mg of alprazolam orally for premedication on the ward.

Patients in the TEA group had their catheter inserted in the anesthetic room with standard monitoring (ECG, non-invasive blood pressure, pulse oximetry). It was inserted by an experienced cardiothoracic anesthesiologist at the T5–T7 level with the patient in the sitting position. An epidural set with an 18-G Tuohy needle (BBraun, Melsungen, Germany) was used with a loss of resistance technique with saline to detect the epidural space. The catheter was inserted 5 cm to the epidural space. A test dose of 3mL of 2% lidocaine with epinephrine in concentration 1:200,000 was administered to exclude the intravascular position of the catheter.

A specialised catheter (PAINfusor, Plan1Health, Amaro, Italy) with a fenestrated length of 7.5 cm was inserted in the SPA group prior to closure of the thoracotomy wound. The thoracic surgeon opened the parietal pleura at the medial pole of the incision and inserted the catheter 10 cm deep to the subpleural paravertebral space under the direct vision through the introducer needle. The catheter was then fixed to the skin.

General anesthesia was performed with intravenous induction using sufentanil, propofol, and rocuronium and maintained with an oxygen/air mixture, desflurane, boluses of sufentanil (according to SPI monitoring), metamizole and rocuronium (train-of-four monitoring, response 1–2 twitches). The patients had their airway secured with a double-lumen Robertshaw tube. All patients were extubated in the operating room, following reversal of muscle relaxation with sugammadex to achieve a train-of-four ratio of greater than 0.95. Immediately after closure of the thoracotomy incision, patients in both groups received a loading dose of 0.25% levobupivacaine via their catheters based on their body weight. The total dose was calculated as follows: 0.2 mL × kg (0.5 mg of levobupivacaine × kg of total body weight). Patients weighing over 100 kg received the same maximum dose of 20 mL (50 mg) of levobupivacaine. All patients were closely monitored by the nursing staff for any neurological or cardiovascular symptoms of local anesthetic systemic toxicity (LAST) during the entire postoperative period. Extent of the local anesthetic block was assessed as per protocol in our hospital by the ice cube dermatomal testing.

During the first two hours after the bolus of levobupivacaine, the only rescue analgesia allowed was in patients with a VAS > 5, who were given piritramide 15 mg s.c. Following the last blood sample (120 min after the bolus), regional analgesia using either a subpleural or epidural catheter was commenced. The two groups (SPA and TEA) received a mixture of 0.2 mcg × mL^−1^ sufentanil with 0.125% levobupivacaine solution through the catheter at a rate of 0.1 mL/kg/h over the 24-h period. In addition, all patients received regular medication of metamizole 1 g (2–10–18 h after loading dose) and paracetamol 1 g (6–14–22 h after the loading dose). Piritramide 15 mg s.c. was allowed as a rescue therapy in patients with a VAS > 4. Absence of improvement of VAS (VAS > 4) despite rescue medication was defined as a failure of the regional anesthesia technique. Hypotension requiring noradrenaline administration was defined as a drop in blood pressure of more than 25% of the baseline value or mean arterial pressure < 65 mmHg.

### 2.3. Outcomes

The primary outcome of the study was the plasma level of levobupivacaine 30 min following the administration. In addition, the levels at 60 and 120 min were also studied and compared. The blood samples were taken from the patient‘s central venous catheter. Pharmacokinetic modeling was performed based on the obtained plasma levels and patient weight, height and creatinine concentration.

Secondary outcomes included: Visual Analogue Score (VAS) on the scale 0−10 comparisons between the groups at 1, 2, 6, 12 and 24 h, mobility of the patients using the AM-PAC “6-click” score (Boston University Functional Assessment Score) [11] (Table 1) at 1, 2, 6, 12 and 24 h, comparison of need for pharmacological support of circulation with norepinephrine at 30 min, 6, 12 and 24 h after the procedure, changes in neurological status and Glasgow Coma Scale. The total success rate of regional anesthesia technique and their complications were also recorded. Recorded complications of regional anesthesia techniques included LAST, local complications (hematoma, infection) and neurological complications such as paresthesia, symptoms of nerve root irritation, urinary retention or weakness of the lower extremities.

### 2.4. Pharmacokinetic Analysis

Levobupivacaine serum concentrations were measured using liquid chromatography with tandem mass spectrometric detection in positive ESI mode (LC-ESI(+)−MS/MS); mepivacaine was used as the internal standard (IS). Internal standard solution (10 µL, c = 125 ng × mL^−1^ in methanol) and 100 µL of borate buffer (pH = 9.0; 0.1 mol × L^−1^) were added to 100 µl of serum sample in a 1.5 mL Eppendorf tube. The solution was shaken for 10 s. Following this, 200 µL of ethyl acetate was added into the mixture and it was vortexed for 30 s. The sample was then centrifuged (9600 g, 90 s) and the supernatant (100 µL) was transferred to insert it into an autosampler vial and it was then evaporated to dryness. The residue was diluted with 100 µL of mobile phase (50:50, 0.1% formic acid in water and acetonitrile).

The method was developed using Nexera X2 Shimadzu HPLC (Nakagyo-ku, Kyoto, Japan) coupled with AB Sciex QTRAP 5500 (MA, USA). Mobile phase A consisted of 0.1% formic acid in water and mobile phase B consisted of acetonitrile. The analysis was performed on Zorbax Eclipse XDB-C18 column (1.8 µm, 50 × 4.6 mm). The initial LC conditions had a flow rate of 0.5 mL × min^−1^ at a mobile phase composition of 50:50 (A:B). These conditions were held for 120 s to load the analytes onto the column. At 120 s the mobile phase composition was ramped to 100 % (B) within 78 s and held for 30 s and then returned to initial LC conditions.

Quantitation was done using multiple reaction monitoring (MRM) mode to monitor protonated precursor—product ion transition of m/z 288.6—140.0 for bupivacaine and 246.6—98.2 for mepivacaine.

Method performance was evaluated following the recommendations of the Scientific Working Group for Forensic Toxicology [12] and it has met the required criteria for all analytes. The calibration range of the assay was 5–1000 ng × mL^−1^ with regression coefficient of 0.9856 in the linear model, LOQ was 5 ng × mL^−1^, bias less than 8.6 %, intra- and inter-day precision was less than 8.6 % (CV).

### 2.5. Pharmacokinetic Modeling

Individual levobupivacaine pharmacokinetic parameters—volume of distribution (V_d_), clearance (CL), elimination half-life (t_1/2_), elimination and absorption rate constant (K_e_ and K_a_), maximum serum concentration (C_max_), time to maximum serum concentration (t_max_) and area under the serum drug concentration-time curve (AUC) were calculated in a one-compartmental pharmacokinetic model with first-order absorption and elimination kinetics based on individual demographic and clinical data, and observed levobupivacaine serum levels using MWPharm^++^ software (MediWare, Prague, Czech Republic). The levobupivacaine population pharmacokinetic model was derived from Simon et al. [13], and then individualized to maximize the fit of the simulated pharmacokinetic profile curve with observed concentration points in each patient (Figure 1). The fit was performed using the Bayesian method.

### 2.6. Statistics

Power analysis was performed prior to commencement of the study. We used published data from the epidural application of levobupivacaine [6]. The standard deviation of the outcome variable (plain levobupivacaine) was 0.15 mg × L^−1^, with a proposed significant difference between the groups of 0.14 mg × L^−1^ (20% of mean). Statistical power was 80% (β = 0.2) and α = 0.05. A sample size of 19 patients per group was calculated. We planned to randomize 44 patients in total to compensate for potential dropouts. The data were first analyzed for normal distribution using the Shapiro-Wilk test.

According to data distribution, ANOVA test with repeated measures was used for comparison of epidural and subpleural concentrations between both branches as well as between the concentrations in particular times after the application. For the significant results, the post-hoc tests with correction were performed. The between-subjects pairwise comparison was tested by the Student *t*-test. Both post-hoc test results were adjusted by Bonferroni correction. The within-subjects factor pairwise comparison was tested within the repeated measure ANOVA model. For the quality of analgesia and comparison of AM-PAC scores, the normality was rejected and Mann Whitney U test with correction was used. The comparison of hemodynamic stability and complications categories was performed by Fisher’s exact test. MedCalc Statistical Software version 19.1.5 (MedCalc Software bv, Ostend, Belgium; https://www.medcalc.org; 2020) was used for all comparisons and *p* levels < 0.05 were considered statistically significant.

## 3. Results

### 3.1. Patient Characteristics

In total 75 patients scheduled for open thoracic procedures were initially screened for eligibility. Fifty of these were enrolled, and forty-four patients were finally analyzed for the primary outcome (Figure 2).

Demographic data between the groups, including gender, age, height, weight, body surface area (BSA), body mass index (BMI) and creatinine plasma concentration were without statistical difference (Table 2).

### 3.2. Plasma Concentrations

Measured concentrations in plasma 30, 60 and 120 min after application are summarized in Table 3. The repeated measure ANOVA results were *p* = 0.032 for the between-subjects effect (difference between the branches) and *p* < 0.0001 for within-subjects effect (Greenhouse-Geiser estimate). Concentrations at 30 min were normally distributed and without statistical difference. They were 0.389 mg × L^−1^ (95% CI 0.330–0.447 mg × L^−1^) in the thoracic epidural group and 0.318 mg × L^−1^ (95% CI 0.257–0.379 mg × L^−1^) in the subpleural analgesia group (*p* = 0.33).

Changes in plasma concentration of local anesthetic in time were similar in both groups—highest concentrations were seen 30 min after application, while they decreased at both 60 and 120 min after levobupivacaine administration, however only at 120 min, the difference in particular branches was significant. The systemic plasma level of levobupivacaine was significantly higher in the thoracic epidural group at 120 min.

Calculated maximum serum concentrations (C_max_) were not significantly higher in the TEA group while the difference in the area under the serum drug concentration-time curve (AUC) was statistically significant. Time to achieve maximum plasma concentration (t_max_) was without any statistical difference between the groups. Dose normalized concentrations of levobupivacaine in both groups are expressed in Figure 3. The reduction of levobupivacaine concentration in time between each consecutive measurement was significant in both branches.

### 3.3. Clinical Secondary Outcomes

These findings are summarized in Table 4. The quality of analgesia was lower in the subpleural paravertebral group at 1 h, and it was without any statistical difference at 2, 6, 12 h. Mobility, as measured with AMP-AC score, was without any difference between TEA and SPA patients. The requirement of systemic analgesia, hemodynamic stability, and incidence of complications were also without statistical difference. Three patients (13.6%) in the TEA group required continuous infusion of noradrenaline (doses 0.01–0.15 µg × kg^−1^ × min^−1^). There was no need for vasopressor support in the SPA group. Four patients (18.2%) experienced complications associated with thoracic epidural analgesia, three of which (13.6%) had temporary neurological deficits which resolved spontaneously.

## 4. Discussion

In this study, we primarily compared the pharmacokinetics of the selective S-enantiomer of bupivacaine, levobupivacaine after bolus administration to the subpleural paravertebral space with administration to the thoracic epidural space. We also modeled peak plasma levels of the drug in time to ensure that they did not reach a threshold for systemic toxicity. Plasma levels of levobupivacaine at which patients show symptoms of systemic toxicity (either neurological or cardiac) have been reported in healthy volunteers as between 2.4 and 2.7 mg × L^−1^ [14,15,16]. Mean plasma levels in our study were more than six times lower than these reported values. The maximum plasma concentration of levobupivacaine did not even reach 1 mg × L^−1^ in either group of patients. No studies have determined toxic levels in elderly patients with cardiovascular or other comorbidities nor any effects of chronic medication on the toxic threshold of levobupivacaine. The maximum levels reported in our study were 0.784 mg × L^−1^ in the subpleural paravertebral block and 0.668 mg × L^−1^ in the epidural analgesia which should be far below any potential toxic levels. No symptoms of cardiovascular or neurological toxicity were seen in our patients.

Several studies have been published on the pharmacokinetics of levobupivacaine in different regional anesthesia techniques in adults. Most of the articles have been focused on the administration of levobupivacaine into the interfascial spaces, such as the transversus abdominis plane, the rectus sheath or fascia iliaca compartment. These were performed on healthy volunteers [17,18] or ASA classification I-II patients indicated for elective surgeries [19]. Yasumura et al. studied the pharmacokinetics of 2.5 mg × L^−1^ levobupivacaine in the transversus abdominis plane and rectus sheath blocks [20]. They found C_max_ similar in both groups with means of approximately 1 mg × L^−1^ and faster absorption from the transversus abdominis plane. One study investigated the pharmacokinetics of levobupivacaine after a single bolus of the drug to the fascia iliaca compartment in elderly frail patients with a fractured neck of the femur [21]. The authors reported the highest C_max_ as 1.42 mg × L^−1^ and that none of their patients exhibited any symptoms of local anesthetic systemic toxicity (LAST). Kopacz et al. studied the pharmacokinetics of 0.5% levobupivacaine administered into the lumbar epidural space in ten patients scheduled for spine surgery [7]. They reported a mean plasma level of 0.36 mg × L^−1^ with no symptoms of LAST. However, the authors did not report the highest plasma concentrations achieved in their cohort. Another study also assessed the plasma concentration of levobupivacaine following administration of 18 mL of 0.75% levobupivacaine into the lumbar epidural space [9]. Perotti et al. studied 181 patients receiving a continuous lumbar epidural infusion of levobupivacaine 0.125% or ropivacaine 0.2% for postoperative pain relief [8]. They found the highest C_max_ in the levobupivacaine group of 2.13 at T_max_ of 48 h which indicates the cummulation of local anesthetics in time. Paravertebral bolus application of 19 mL of 0.25% levobupivacaine resulted in a mean maximum plasma concentration of 0.51 mg × L^−1^, with the highest plasma concentration of 0.881 mg × L^−1^ and no patients exhibited any symptoms of LAST [10]. These findings are similar to our plasma levels in the SPA group. The highest plasma levels found in the literature were following the axillary approach to the brachial plexus—3.74 mg × L^−1^ [22] and combined psoas compartment and sciatic nerve block—3.1 mg × L^−1^ [23], respectively.

This study has several pharmacokinetic and clinical limitations. We only measured total plasma concentrations of levobupivacaine. It has been suggested that a measurement of free unbound levobupivacaine concentration can be more accurate for the prediction of LAST. Another limitation is that we did not measure the plasma concentration of levobupivacaine after continuous administration for a 24 h interval. This could provide useful information about possible local anesthetic cumulation in the body. This study was only powered for the primary outcome which was a difference in the plasmatic levels of levobupivacaine at 30 min after its administration. Therefore we cannot prove any statistical significance in the secondary outcomes or peak plasmatic concentration of the local anesthetic drug. Main clinical limitation of the study is that the doses of levobupivacaine routinely administered into the epidural catheter are significantly lower than those in our study. We adjusted the doses to achieve an equivalent amount in both groups. A relative “overdose” in the thoracic epidural group probably contributed to the recorded episodes of hypotension or temporary neurological deficit in this group.

## 5. Conclusions

Both methods of regional analgesia used in this study showed significantly lower plasma concentrations of levobupivacaine after bolus application at the end of the surgical procedure than the published toxic levels of this local anesthetic. We did not report any symptoms of LAST and therefore we consider 0.25% levobupivacaine in this calculated dosage as a safe technique for continuous thoracic epidural analgesia and subpleural paravertebral analgesia. Continuous subpleural paravertebral analgesia may also be considered as an alternative to thoracic epidural catheters in patients in whom this technique is contraindicated.

## Figures and Tables

**Figure 1 jcm-09-01395-f001:**
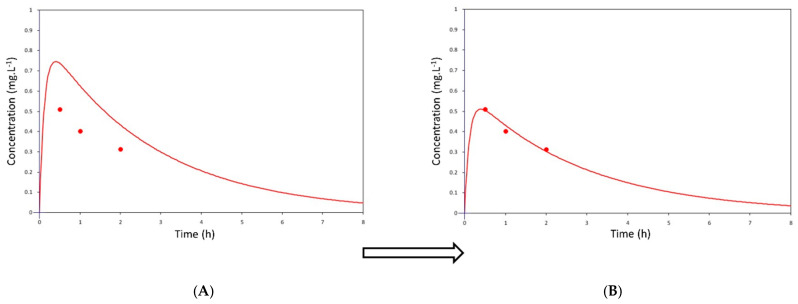
The methodology of pharmacokinetic modeling. Bayesian approach defines all unknown parameters as random variables and via a large number of subsequent iterations the variables are adapted taking into account the physiological and substance properties to achieve maximal fitting of the simulated pharmacokinetic profile curve with the real measured concentration points in each patient. “A priori” concentration-time levobupivacaine population pharmacokinetic profile derived from previously published data [12] and real measured levobupivacaine concentrations in a representative patient from our cohort (**A**). “A posteriori” concentration-time levobupivacaine pharmacokinetic profile is individualized to maximize the fit of the simulated pharmacokinetic profile curve with the observed concentration points in the representative patient (**B**).

**Figure 2 jcm-09-01395-f002:**
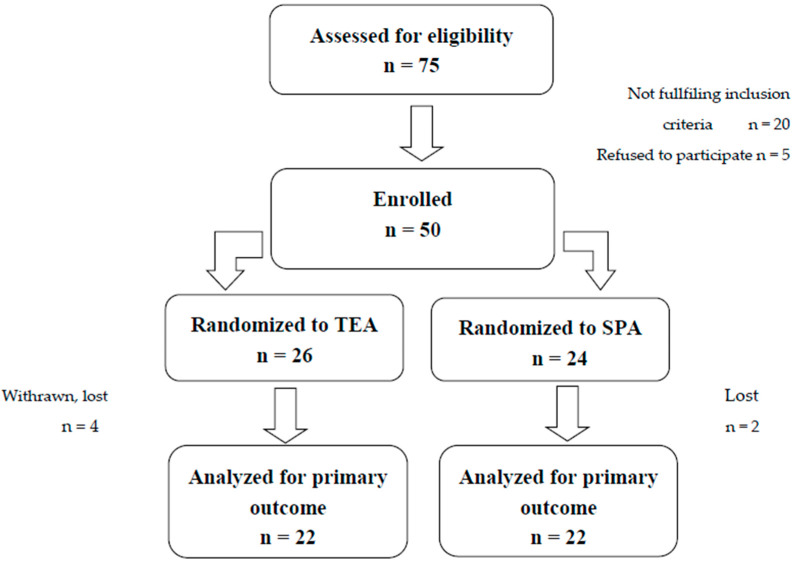
CONSORT flow diagram of the study. Six patients were excluded from the final analysis; TEA group—two patients withdrawn from the study, one patient lost due to an artificial removal of the catheter in the operating room, one patient excluded due to a lost blood sample. SPA group—one patient excluded due to a human error in sampling, one patient excluded because of the insufficient blood sample.

**Figure 3 jcm-09-01395-f003:**
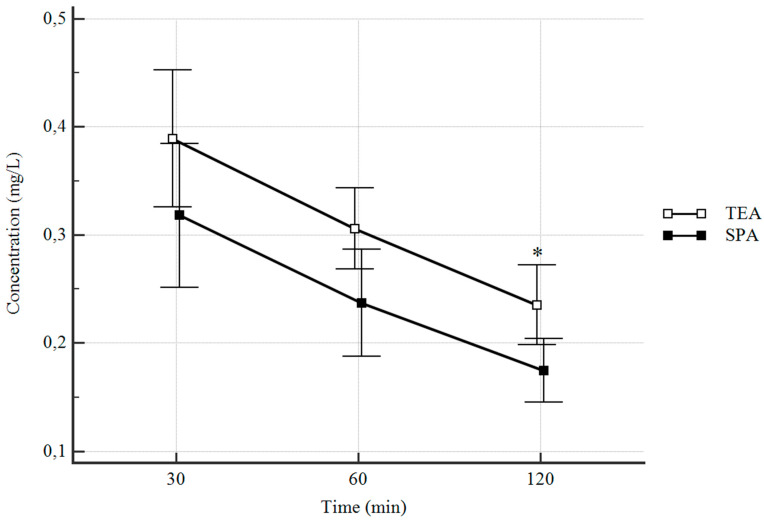
Levobupivacaine serum concentration-time course after epidural and subpleural paravertebral administration. Data are presented as mean (95% CI). * Significantly different (*p* < 0.05).

**Table 1 jcm-09-01395-t001:** Boston University AM-PAC “6-clicks” Basic Mobility Inpatient Short Form.

How Much Difficulty Does the Patient Currently Have?	Unable	A Lot	A Little	None
Turning over in bed (including adjusting bedclothes, sheets and blankets)?	1	2	3	4
Sitting down on and standing up from a chair with arms (e.g., wheelchair, bedside commode, etc.)?	1	2	3	4
Moving from lying on back to sitting on the side of the bed?	1	2	3	4
**How much help from another person does the patient currently need?**	**Total**	**A lot**	**A little**	**None**
Moving to and from a bed to a chair (including a wheelchair)?	1	2	3	4
To walk in hospital room?	1	2	3	4
Climbing 3–5 steps with a railing?	1	2	3	4

**Table 2 jcm-09-01395-t002:** Characteristics of the groups, demographic data.

	Thoracic Epidural Analgesia (TEA)	Subpleural Paravertebral Analgesia (SPA)
	(*n* = 22)	(*n* = 22)
Gender (M/F)	10/12	11/11
Age (years)	72.5 [48–80]	68 [53–84]
Height (cm)	168 [154–187]	172 [159–188]
Weight (kg)	74 [51–116]	85.5 [48–119]
Body surface area (BSA)	1.81 [1.54–2.37]	1.98 [1.47–2.44]
Body mass index (BMI)	25.5 [18.1–39.7]	28.9 [19–38.1]
Creatinine (µmol × L^−1^)	70 [53–96]	71 [42–144]

Data expressed as median [range].

**Table 3 jcm-09-01395-t003:** Pharmacokinetic parameters.

	TEA (*n* = 22)	SPA (*n* = 22)	*p*
Plasma concentration at 30 min	0.389 (0.326–0.452)	0.318 (0.252–0.385)	0.33 *
	[0.09–0.668]	[0.122–0.784]	
Plasma concentration at 60 min	0.306 (0.268–0.34)	0.237 (0.188–0.288)	0.08 *
	[0.101–0.449]	[0.99–0.559]	
Plasma concentration at 120 min	0.235 (0.198–0.272)	0.175 (0.145–0.204)	0.03 *
	[0.083–0.421]	[0.09–0.314]	
C_max_	0.396 (0.341–0.451)	0.320 (0.256–0.384)	0.069
	[0.107–0.605]	[0.149–0.777]	
t_max_	0.46 (0.39–0.53)	0.40 (0.36–0.44)	0.17
	[0.318–0.86]	[0.313–0.803]	
AUC	1.262 (1.04–1.48)	0.906 (0.74–1.07)	0.039
	[0.427–2.539]	[0.418–1.666]	

Data expressed as means (95% CI) [range]. Plasma concentration (mg × L^−1^), C_max_ (mg × L^−1^), t_max_ (h), AUC (mg × h × L^−1^); * Bonferoni correction of the *p*-value.

**Table 4 jcm-09-01395-t004:** Pain scores, mobility, hemodynamic stability, complications.

		TEA (*n* = 22)	SPA (*n* = 22)	*p*
Nurse rating scale (NRS)	1 h	3 [1–5]	5 [3–6.25]	0.036
	2 h	3 [1–5.25]	4 [3–5]	0.44
	6 h	3 [1.75–4]	4 [2.75–4]	0.064
	12 h	2 [1–3]	3 [2–4]	0.064
	24 h	2 [1–3]	3 [2–4]	0.061
AMP–AC	1 h	6 [6–6]	6 [6–6]	0.99
	2 h	6 [6–7]	6 [6–7]	0.307
	6 h	7 [6.75–8]	7 [6.75–8]	0.936
	12 h	8 [7–9.25]	8 [7–9]	0.96
	24 h	14 [11.5–15.25]	12 [10.75–15.25]	0.603
Hemodynamic instability		3 (13.6%)	0	0.232
Complications - Local hematoma - Temporary paresthesia - Transient weakness of leg(s)		4 (18.2%) 1 (4.5%) 1 (4.5%) 2 (9.1%)	0	0.108

Data expressed as median [25–75 interquartile range] or total numbers (%).

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
