# Peer review of "A Randomized Comparison of Plasma Levobupivacaine Concentrations Following Thoracic Epidural Analgesia and Subpleural Paravertebral Analgesia in Open Thoracic Surgery"

_jcm, 2020, doi:10.3390/jcm9051395_

Round 1

Reviewer 1 Report

Thank you for requesting my opinion for the review of the manuscript "A randomized comparison of plasma levobupivacaine concentrations following thoracic epidural analgesia and subpleural paravertebral analgesia in open thoracic surgery"

The manuscript is well written, causes little confusion.

The research question is well developed.

No major modification related to statistics.

I suggest avoiding a statistical test to compare patient demographics, a comparison not recommended for randomized controlled trials, and replace it with the standardized difference.

Some minor grammatical corrections are to be made.

I hope this will help.

Issam.

Author Response

I suggest avoiding a statistical test to compare patient demographics, a comparison not recommended for randomized controlled trials, and replace it with the standardized difference.

Response: statistical tests to compare demographic data of the patients were erased.

Reviewer 2 Report

The authors present an interesting randomised controlled trial examining the relative pharmacokinetics of levobupivacaine when administered via either the thoracic epidural or paravertebral route.

The narrative is well written, and it is a significant strength that the investigation is a robustly conducted randomised controlled trial; a methodology that might not be considered essential given the pharmacokinetic primary outcome.

Major comments:

Clinical relevance

Whilst the methodological design of the study (giving an identical dose of local anaesthetic via two different routes immediately after wound closure) is an elegant method for comparing the relative pharmacokinetics of the epidural and paravertebral spaces, my major concern with this study is that this is widely divergent from routine clinical practice.

Firstly, in my experience it is very unusual to provide similar volumes of local anaesthetic to the epidural and paravertebral spaces; in our institution the volume routinely administered to the epidural space is roughly a third of that administered paravertebrally (both in terms of initial bolus and infusion rate). This may account for the significant incidence of hypotension and ‘temporary neurological deficit’ in the current study. Temporary neurological deficit is insufficiently described / discussed in the narrative; the conclusion of the ‘safety’ of the technique requires some justification given the 13.6% incidence of neurological deficit.

Secondly, administering the first bolus of epidural analgesia at the end of anaesthesia rather than prior to knife to skin is unusual. Indeed, the protocol might be considered to be depriving the thoracic epidural group of any potential benefit which could have be derived intraoperatively in ordinary circumstances (e.g. reduced chronic pain, less opiate administration, haemodynamic stability).

Nomenclature

The authors use the terms ‘subpleural paravertebral’, ‘paravertebral’ and ‘subpleural’ interchangeably when describing the non-epidural group. Consistency throughout the manuscript desirable. Further, given the wide variation in clinical practice as to what is considered a ‘subpleural’ or ‘paravertebral’ block further detail is needed describing the specific method of the block. On page 2, line 49, ‘continuous paravertebral block’ and ‘subpleural analgesia’ are listed as distinct, alternative techniques rather than pseudonyms for the same technique.

Blinding

The description of the blinding methodology is inconsistent through the manuscript. The statement page 2, line 79: “Only patients were blinded to group allocation”, appears in conflict with the statement Page 3, line 117: “The member of the medical staff who monitored VAS scores, side effects, and rescue analgesia requirements was blinded to the study.”

Over interpretation

Page 9, line 273: “Our study also showed that subpleural analgesia using levobupivacaine is not significantly inferior to thoracic epidural analgesia in regards to pain intensity or the need for the additional parenteral application of strong opioids.” Whilst the study sample size and methodology appear entirely appropriate to assess the primary outcome, this statement concerning secondary outcomes is far too strong. This study was not designed to, nor is it sufficiently powered to conclude ‘non-inferiority’ between the two techniques.

PK modelling

My specialist area of knowledge is thoracic anaesthesia rather than pharmacokinetics (and therefore I make no attempt to critically review the PK methodology), but I feel it would aid understanding / readability for the anaesthetic (non-pharmacokinetically trained) audience if indicative patient results, and pharmacokinetic models could be presented in diagrammatic / graphical form illustrating how results for Cmax, tmax and AUC were derived). It is not clear to me for example how tmax values can be derived from 3 data points per patient? To what extent might the limited number of data points per patient influence the broadness of the confidence intervals (i.e. are these broad due to limited data or inter-patient variability)?

Minor comments

Timing of local anaesthetic injection – described as “before the thoracotomy closure” in the abstract, and “immediately after closure” in the methods!?

Page 3, line 105: The acronym ‘LAST’ requires to be defined at first use.

Page 4, line 153: coefficients of variation or similar should be provided to support the ‘method performance’ of the analytical technique.

Page 4, line 169: In the power analysis, it appears that a significant difference between groups of 20% was sought based on this difference having been observed in a comparison of epidural levobupivacaine with and without epinephrine. This inference is not immediately logical and requires further justification.

Page5, Results: How were the 6 patients randomised but not analysed ‘lost’?

Was any clinical assessment made of local anaesthetic blockade e.g. by dermatome mapping?

Table 4. I am not convinced the data nor statistical comparisons displayed within table 4 add anything to the manuscript when the same data can be easily visualised in the immediately adjacent figure 2, and in table 3. Further, comparisons of local anaesthetic concentration within groups are of limited relevance to the aims of the study and simply demonstrate that following an initial bolus of local anaesthetic, plasma concentrations fall over time.

Table 5. it is not clear what is being referred to by the term ‘complications’ in table 5. Perhaps the definition of this composite (which also requires to be defined in the method section) could be added to the footnote?

Author Response

The authors present an interesting randomised controlled trial examining the relative pharmacokinetics of levobupivacaine when administered via either the thoracic epidural or paravertebral route.

The narrative is well written, and it is a significant strength that the investigation is a robustly conducted randomised controlled trial; a methodology that might not be considered essential given the pharmacokinetic primary outcome.

Major comments:

Clinical relevance

Whilst the methodological design of the study (giving an identical dose of local anaesthetic via two different routes immediately after wound closure) is an elegant method for comparing the relative pharmacokinetics of the epidural and paravertebral spaces, my major concern with this study is that this is widely divergent from routine clinical practice.

Response: we agree with the reviewer that the doses of local anesthetic solutions used in this study differ from the doses used in routine clinical practice. In order to achieve equivalent doses for both methods used, we used increased bolus dose to the thoracic epidural catheter and slightly decreased dose into subpleural paravertebral. We mention this in a revised version of the manuscript in the „Discussion“ section.

"Another limitation is that in routine clinical settings the doses of local anesthetic solutions administered into the epidural catheter are significantly lower than in subpleural paravertebral analgesia. We adjusted the doses to achieve an equivalent amount in both groups.“ 

Firstly, in my experience it is very unusual to provide similar volumes of local anaesthetic to the epidural and paravertebral spaces; in our institution the volume routinely administered to the epidural space is roughly a third of that administered paravertebrally (both in terms of initial bolus and infusion rate). This may account for the significant incidence of hypotension and ‘temporary neurological deficit’ in the current study. Temporary neurological deficit is insufficiently described / discussed in the narrative; the conclusion of the ‘safety’ of the technique requires some justification given the 13.6% incidence of neurological deficit.

Response:

We agree that the volumes administered to the epidural and paravertebral spaces are different in clinical practice. In our institution, the volumes also differ, but not as significantly as in the reviewer´s institution. We apply routinely about 60-70% of the epidural dose in comparison with the subpleural paravertebral application. We feel that with the lateral thoracotomy incision, maximum of three dermatomes (the thoracotomy level and two adjacent ones) should be covered by analgesia and thus excessive doses of local anesthetic solution are not required.

Secondly, administering the first bolus of epidural analgesia at the end of anaesthesia rather than prior to knife to skin is unusual. Indeed, the protocol might be considered to be depriving the thoracic epidural group of any potential benefit which could have be derived intraoperatively in ordinary circumstances (e.g. reduced chronic pain, less opiate administration, haemodynamic stability).

Response: Clinical practice differs widely among institutions. While some centers activate epidural catheter prior to intubation, other centers prior to skin incision, others use the first bolus of the local anesthetic solution just before wound closure or extubation (Swenson BR, Gottschalk A, Wells LT, et al. Intravenous lidocaine is as effective as epidural bupivacaine in reducing ileus duration, hospital stay, and pain after open colon resection: a randomized clinical trial. Reg Anesth Pain Med 2010; 35: 370–6; Mukherjee M, Goswami A, Gupta SD et al. Analgesia in post-thoracotomy patients: comparison between thoracic epidural and thoracic paravertebral blocks. Anesth Essays Res 2010; 4: 75-80; Strandby RB, Ambrus R, Achiam MP et al. Effect of early versus delayed activation of thoracic epidural anesthesia on plasma pro-atrial natriuretic peptide to indicate deviations in central blood volume during esophagectomy. Reg Anesth Pain Med 2019; doi: 10.1136/rapm-2019-100508. ). In our center and also in other centers in continental Europe, the catheter is usually activated before the skin closure because we worry of a combination of the hypotensive effect of GA and epidural anesthesia. In this study, we also chose identical timing of catheter activation in both groups in order to similarize the pharmacokinetic conditions between the groups.

Nomenclature

The authors use the terms ‘subpleural paravertebral’, ‘paravertebral’ and ‘subpleural’ interchangeably when describing the non-epidural group. Consistency throughout the manuscript desirable. Further, given the wide variation in clinical practice as to what is considered a ‘subpleural’ or ‘paravertebral’ block further detail is needed describing the specific method of the block. On page 2, line 49, ‘continuous paravertebral block’ and ‘subpleural analgesia’ are listed as distinct, alternative techniques rather than pseudonyms for the same technique.

Response: thank you for this comment. The terms to describe surgically inserted catheters in the literature, as well as their methodology, are inconsistent in the literature. We unified the nomenclature throughout to manuscript to „subpleural paravertebral“.

We also understand that the term „paravertebral analgesia“ or „continuous paravertebral block“ is reserved for percutaneous (landmark-based or ultrasound-guided) technique while „subpleural paravertebral“ refers to the surgically inserted catheter after the opening of parietal pleura medially to the paravertebral space – as described in Methods section – page 3, lines 97-101.

We also changed the description of regional anesthesia techniques used for thoracic surgery as follows:

“Other methods of regional analgesia, such as continuous percutaneous paravertebral block, subpleural paravertebral analgesia,  interpleural administration or continuous erector spinae block have been proposed as alternatives [4].”

Blinding

The description of the blinding methodology is inconsistent through the manuscript. The statement page 2, line 79: “Only patients were blinded to group allocation”, appears in conflict with the statement Page 3, line 117: “The member of the medical staff who monitored VAS scores, side effects, and rescue analgesia requirements was blinded to the study.”

Response: We understand the reviewer´s comment. – The statement page 2, line 79 „Only patients were blinded to group allocation“ means that the anesthesiologists and surgeons performing the procedures and filling the forms in the operating room were not blinded – this was not possible because they inserted the thoracic epidural catheters/subpleural paravertebral catheters. We changed the sentence to avoid any confusion:

„The patients were blinded to group allocation. The blinding was not feasible for the anesthesiologists or surgeons performing the procedures in the operating room.“  

Over interpretation

Page 9, line 273: “Our study also showed that subpleural analgesia using levobupivacaine is not significantly inferior to thoracic epidural analgesia in regards to pain intensity or the need for the additional parenteral application of strong opioids.” Whilst the study sample size and methodology appear entirely appropriate to assess the primary outcome, this statement concerning secondary outcomes is far too strong. This study was not designed to, nor is it sufficiently powered to conclude ‘non-inferiority’ between the two techniques.

Response: We agree with this comment. We commented on the results of secondary outcomes and their limitation also in the „Discussion“ section of the manuscript.  The statement was softened to: „in this limited number of patients, subpleural paravertebral analgesia showed similar pain relief as thoracic epidural analgesia.

PK modelling

My specialist area of knowledge is thoracic anaesthesia rather than pharmacokinetics (and therefore I make no attempt to critically review the PK methodology), but I feel it would aid understanding / readability for the anaesthetic (non-pharmacokinetically trained) audience if indicative patient results, and pharmacokinetic models could be presented in diagrammatic / graphical form illustrating how results for Cmax, tmax and AUC were derived). It is not clear to me for example how tmax values can be derived from 3 data points per patient? To what extent might the limited number of data points per patient influence the broadness of the confidence intervals (i.e. are these broad due to limited data or inter-patient variability)?

Response:  Thank you for this comment. Indeed, the limited number of points does not allow direct calculations of the pharmacokinetic parameters for the individual patients. However, a method considered to be a gold standard to deal with the issue of a limited number of drug levels in both clinical studies as well as in routine therapeutic drug monitoring (where usually the initial pharmacokinetics is calculated based on 1-2 samples only) is to estimate the individual PK parameters based on a pharmacokinetic modeling approach that utilizes Bayesian statistics (1). In a brief and simple description, a population pharmacokinetic model is constructed based on available bibliographic data. Subsequently, Bayesian approach defines all unknown parameters as random variables and via a large number of subsequent itinerations (e.g. 106) the variables are adapted taking into account the physiological and substance properties in order to achieve maximum fitting of the simulated pharmacokinetic profile curve with the truth observed concentration points in each patient. The result is PK profile that fits the 3 truth drug levels of each individual patient, from which PK characteristics may be derived including for e.g. a Tmax value at a time, where no real sample has been taken.

The variability is in that case primarily driven by a number of patients and interpatient variability. The number of samples in each of the patients is a driving factor for the precision of the model, while 3 samples per patient are usually considered a minimum adequate number to predict the PK profile as a whole (2). Therefore it should not be the key factor in our study to determine large variability.

Although we could show initial and final model PK parameters, this would not be of much value for an aexplanation of principles of Bayessian PK modeling. Moreover, individual PK profile was simulated for each patient. Therefore, 44 profiles would have to be shown.

(1) Fuchs A et al. Benchmarking therapeutic drug monitoring software: a review of available computer tools. Clin Pharmacokinet 2013;52:9–22. (2) David O et al. Limited sampling strategies. Clin Pharmacokinet 2000;39(4):311-3.

Minor comments

Timing of local anaesthetic injection – described as “before the thoracotomy closure” in the abstract, and “immediately after closure” in the methods!?

Response: we apologize. There is a mistake. The catheter was inserted by the surgeon under direct vision before the thoracotomy closure but the bolus of local anesthetic was given immediately after the closure. Corrected in the manuscript.

Page 3, line 105: The acronym ‘LAST’ requires to be defined at first use.

Response: „local anesthetic systemic toxicity“ (LAST) – corrected in the manuscript

Page 4, line 153: coefficients of variation or similar should be provided to support the ‘method performance’ of the analytical technique.

Response: added to the manuscript

„The calibration range of the assay was 5–1000 ng.ml-1 with regression coefficient of 0.9856 in the linear model, LOQ was 5 ng.ml-1, bias less than 8.6 %, intra- and inter-day precision was less than 8.6 % (CV).“

Page 4, line 169: In the power analysis, it appears that a significant difference between groups of 20% was sought based on this difference having been observed in a comparison of epidural levobupivacaine with and without epinephrine. This inference is not immediately logical and requires further justification.

Response:  this is very difficult to find any relevant data on the concentrations of local anesthetic solution, namely levobupivacaine, at the thoracic level. The only study available was that of Kopacz et al. (we used it for our sample size calculation).

Only the concentrations of levobupivacaine without any additive was used as a starting point –

The proposed difference of 20% was considered as clinically significant. All pharmacokinetic studies published on levobupivacaine has a limited number of patients (samples) due to the financial expensiveness of the laboratory methods (usually 6-30 patients). 

Page5, Results: How were the 6 patients randomised but not analysed ‘lost’?

Response:

6 patients lost – 4 in the thoracic epidural group, 2 in the subpleural paravertebral group.

2 patients in the thoracic epidural group decided to withdraw from the study

1 patient – in the thoracic epidural group - artificial removal of the catheter in the operating room during moving the patient from the operating table to the bed, 1 patient in the thoracic epidural group – blood sample at 30 mins lost.

1 patient in the subpleural paravertebral group – blood sample at 60 mins not performed (nursing error), 1 patient in the subpleural paravertebral group – blood sample at 60 mins reported by the lab as insufficient.   

Was any clinical assessment made of local anaesthetic blockade e.g. by dermatome mapping?

Response: Extent of the local anesthetic block was assessed as per protocol in our hospital by the assessment of dermatomal analgesia (ice cube dermatomal testing). This was added to the body of the revised manuscript – Method section

Extent of the local anesthetic block was assessed as per protocol in our hospital by the ice cube dermatomal testing.   

Table 4. I am not convinced the data nor statistical comparisons displayed within table 4 add anything to the manuscript when the same data can be easily visualised in the immediately adjacent figure 2, and in table 3. Further, comparisons of local anaesthetic concentration within groups are of limited relevance to the aims of the study and simply demonstrate that following an initial bolus of local anaesthetic, plasma concentrations fall over time.

Response: We agree. Table 4 has been erased.

Table 5. it is not clear what is being referred to by the term ‘complications’ in table 5. Perhaps the definition of this composite (which also requires to be defined in the method section) could be added to the footnote?

Response: Table 5 was changed to Table 4 (original Table 4 erased).

The definition of complications was added to the body of the revised manuscript – „Methods“ section.

“Recorded complications of regional anesthesia techniques included LAST, local complications (hematoma, infection) and neurological complications such as paresthesia, symptoms of nerve root irritation, urinary retention or weakness of the lower extremities.”

Added to Table 4:

Complications

Local hematoma

Temporary paresthesia

Transient weakness of leg(s)

4 (18.2%)

1 (4.5%)

1 (4.5%)

2 (9.1%)

Round 2

Reviewer 2 Report

Many thanks for the opportunity to review the revised submission of this interesting randomised controlled trial examining the relative pharmacokinetics of levobupivacaine when administered via either the thoracic epidural or paravertebral route.

There are just three changes highlighted in red within the revised manuscript, though on reading the authors’ responses the suggestion that many more changes have been made. This makes reviewing the revised document a frustrating and challenging procedure! Nonetheless, it appears the authors have made multiple changes to the manuscript, and the re-submission is a significant improvement.

My main concern however remains the lack of clinical applicability given that identical does of local anaesthetic are administered by the epidural and paravertebral route (which is a significant deviation from clinical practice). The authors have added an additional limitations statement to the effect, but it nonetheless remains a major limitation of the study. I suspect it is the relative epidural ‘overdose’ which has resulted in the significant incidence of hypotension and temporary neurological deficit in the epidural group; something the authors have neglected to address in their responses.

Randomization and blinding - The use of blinded assessors to assess VAS, mobility and complications is a strength of the study design and should be recorded in section 2.1.

I thank the authors for the extended discussion of their methodology in their responses, but I continue to feel it would aid understanding / readability for the non-pharmacokinetically trained audience if further explanation was offered, preferably in diagrammatic from within the manuscript.

Six patients are described as ‘lost from the analysis’. The causes of loss from the study (reported in the author’s responses to the previous review), should be detailed in the manuscript. Perhaps as a footnote in Figure 1?

Author Response

Dear reviewer, thank you for your valuable comments and continuing effort to improve the quality of our manuscript.

Many thanks for the opportunity to review the revised submission of this interesting randomised controlled trial examining the relative pharmacokinetics of levobupivacaine when administered via either the thoracic epidural or paravertebral route.

There are just three changes highlighted in red within the revised manuscript, though on reading the authors’ responses the suggestion that many more changes have been made. This makes reviewing the revised document a frustrating and challenging procedure! Nonetheless, it appears the authors have made multiple changes to the manuscript, and the re-submission is a significant improvement.

R: We apologize for this. We used "track the changes" function as recommended in our revised manuscript. Surprisingly, only a very limited number of changes made has appeared in red color. In this last revision, we also highlighted all requested changes in red in order to help the reviewer to track the changes better.

My main concern however remains the lack of clinical applicability given that identical does of local anaesthetic are administered by the epidural and paravertebral route (which is a significant deviation from clinical practice). The authors have added an additional limitations statement to the effect, but it nonetheless remains a major limitation of the study. I suspect it is the relative epidural ‘overdose’ which has resulted in the significant incidence of hypotension and temporary neurological deficit in the epidural group; something the authors have neglected to address in their responses.

R: We adressed all these point in this revised version of our manuscript, Discussion section.

"Main clinical limitation of the study is that the doses of levobupivacaine routinely administered into the epidural catheter are significantly lower than those in our study. We adjusted the doses to achieve an equivalent amount in both groups. A relative “overdose” in the thoracic epidural group probably contributed to the recorded episodes of hypotension or temporary neurological deficit in this group.“  

Randomization and blinding - The use of blinded assessors to assess VAS, mobility and complications is a strength of the study design and should be recorded in section 2.1.

R: we made the changes in Section 2.1: " The patients and the assessors of postoperative variables (intensity of pain, mobility, and complications) were blinded to group allocation. The blinding was not feasible for the anesthesiologists or surgeons performing the procedures in the operating room."

I thank the authors for the extended discussion of their methodology in their responses, but I continue to feel it would aid understanding / readability for the non-pharmacokinetically trained audience if further explanation was offered, preferably in diagrammatic from within the manuscript.

R: The methodology of pharmacokinetic modelling according to Bayesian methodology was added to the manuscript - a novel Fig. 1:

Figure 1. The methodology of pharmacokinetic modeling. Bayesian approach defines all unknown parameters as random variables and via a large number of subsequent iterations the variables are adapted taking into account the physiological and substance properties to achieve maximal fitting of the simulated pharmacokinetic profile curve with the real measured concentration points in each patient.

“A priori” concentration-time levobupivacaine population pharmacokinetic profile derived from previously published data [12] and real measured levobupivacaine concentrations in in a representative patient from our cohort (on the left)

“A posteriori” concentration-time levobupivacaine pharmacokinetic profile is individualized to maximize the fit of the simulated pharmacokinetic profile curve with the observed concentration points in the representative patient (on the right).

Six patients are described as ‘lost from the analysis’. The causes of loss from the study (reported in the author’s responses to the previous review), should be detailed in the manuscript. Perhaps as a footnote in Figure 1?

R: we agree. Causes of loss from the study are detailed in the footnote of Fig.2: "Figure 2. CONSORT flow diagram of the study. Six patients were excluded from the final analysis; TEA group – two patients withdrawn from the study, one patient lost due to an artificial removal of the catheter in the operating room, one patient excluded due to a lost blood sample. SPA group – one patient excluded due to a human error in sampling, one patient excluded because of the insufficient blood sample."